# Maternal Resveratrol Supplementation Attenuates Prenatal Stress Impacts on Anxiety- and Depressive-like Behaviors by Regulating *Bdnf* Transcripts Expression in the Brains of Adult Male Offspring Rats

**DOI:** 10.3390/brainsci15020210

**Published:** 2025-02-19

**Authors:** Gerardo Vera-Juárez, Edgar Ricardo Vázquez-Martínez, Raquel Gómez-Pliego, Margarita López-Martínez, Judith Espinosa-Raya

**Affiliations:** 1Laboratorio Multidisciplinario en Ciencias Biomédicas, Escuela Superior de Medicina, Instituto Politécnico Nacional, Mexico City 11340, Mexico; ge.vera1900@gmail.com; 2Departamento de Fisiología y Desarrollo Celular, Instituto Nacional de Perinatología, Mexico City 11000, Mexico; 3Unidad de Investigación en Reproducción Humana, Instituto Nacional de Perinatología-Facultad de Química, Universidad Nacional Autónoma de México, Mexico City 11000, Mexico; vamer@comunidad.unam.mx; 4Sección de Ciencias de la Salud Humana, Departamento de Ciencias Biológicas, Facultad de Estudios Superiores Cuautitlán, Universidad Nacional Autónoma de México, Cuautitlán Izcalli 54740, Mexico; pliegoraquel@comunidad.unam.mx

**Keywords:** prenatal stress, depression, anxiety, resveratrol, brain-derived neurotrophic factor

## Abstract

Background: Prenatal stress has been reported to harm the physiological and biochemical functions of the brain of the offspring, potentially resulting in anxiety- and depression-like behaviors later in life. Trans-Resveratrol (RESV) is known for its anti-inflammatory, anxiolytic, and antidepressant properties. However, whether administering RESV during pregnancy can counteract the anxiety- and depression-like behaviors induced by maternal stress is unknown. Objective: This study aimed to assess the protective potential of RESV against molecular and behavioral changes induced by prenatal stress. Methods: During pregnancy, the dams received 50 mg/kg BW/day of RESV orally. They underwent a movement restriction for forty-five minutes, three times a day, in addition to being exposed to artificial light 24 h before delivery. The male offspring were left undisturbed until early adulthood, at which point they underwent behavioral assessments, including the open field test, elevated plus maze, and forced swim test. Subsequently, they were euthanized, and the hippocampus and prefrontal cortex were extracted for RT-qPCR analysis to measure *Bdnf* mRNA expression. Results: By weaning, results showed that prenatal stress led to reduced weight gain and, in adulthood, increased anxiety- and depression-like behaviors and changes in *Bdnf* mRNA expression. However, these effects were attenuated by maternal RESV supplementation. Conclusions: The findings suggest that RESV can prevent anxiety- and depression-like behaviors induced by prenatal stress by modulating *Bdnf* mRNA expression.

## 1. Introduction

Stress plays a key role throughout the lifetime of a being, but its impact is significant in critical developmental phases [1]. A large amount of evidence emphasizes the impact of psychological stress during pregnancy on various aspects of offspring growth and development [2]. The prenatal period is crucial as it determines neural plasticity and regulates brain programming. Disruptions in these processes can lead to neurological disorders in offspring, including generalized anxiety disorder, depressive disorders, attention deficit hyperactivity disorder, autism spectrum disorders, and schizophrenia [3].

The underlying mechanisms by which prenatal stress (PS) affects the neurodevelopmental programming process remain unclear. However, altered hypothalamic–pituitary–adrenal (HPA) axis [4], reactive oxygen species [5], neuroinflammatory pathways [6], and immune system hyperactivation are believed to play a role [7]. Likewise, these effects are influenced by the type and severity of stress, the timing and duration of exposure, and the gender of the offspring [8]. These factors affect the transcription of essential genes involved in neurodevelopment, such as brain-derived neurotrophic factor (BDNF) [9]. BDNF is the most abundant neurotrophin in the brain. It is crucial for neuronal survival and growth, synaptic transmission, neurogenesis, modulation of neurotransmitters, as well as learning, memory, and stress resistance [10]. Studies suggest that stress during early developmental stages can alter the methylation patterns of the *Bdnf* gene on different brain regions of the offspring, causing a decrease in their expression in the hippocampus and prefrontal cortex [11], which has been implicated in various psychological and neurological disorders, such as anxiety, depression, schizophrenia, Parkinson’s, and Alzheimer’s disease [12,13]. Even though a reduction in total *Bdnf* mRNA and protein is the most often seen effect of PS on BDNF, contradictory results have also been found [9].

Resveratrol (trans-3,5,4′-trihydroxy-trans-stilbene, RESV), a polyphenol found in grapes and berries [14], has a variety of biological properties, including antioxidant, anti-inflammatory, anticancerogenic, and neuroprotective effects [15]. Studies have reported that RESV exerts anxiolytic- and antidepressant-like effects in rodents by downregulating hyperactivity of the HPA axis and upregulating BDNF levels in the hippocampus, amygdala, and prefrontal cortex [16,17]. Similarly, RESV can ameliorate neuroinflammation induced by maternal separation via activating the Sirt1/NF-κB pathway [18]. Considering that this information suggests that RESV reduces the effect of stress, it is intriguing that the effect of maternal RESV supplementation on anxiety-like and depression-like behaviors in the offspring has not been explored.

In the present study, the male offspring rat was used to investigate whether maternal resveratrol intake can prevent prenatal stress-induced anxiety and depression-like behaviors in adulthood and, if so, whether the mechanism involves the modulation of the *Bdnf* expression in the hippocampus and prefrontal cortex.

## 2. Materials and Methods

### 2.1. Drugs

RESV was purchased from Sigma Chemical Co. (St. Louis, MO, USA). The drugs were dissolved in an aqueous solution of 0.5% sodium carboxymethyl cellulose, and the vehicle (VEH) groups received only this solution.

RSV treatment: The daily dose was 50 mg/kg BW. Using a previously defined formula [19], 50 mg/kg BW in the rat would correspond to 8.108 mg/kg in humans. This daily dose is within the range used in human studies in which the actions of RESV on several inflammatory and metabolic markers were investigated [20].

### 2.2. Animals

Twenty-four female Wistar rats (200 and 250 g) were provided from the Instituto Nacional de Perinatología Isidro Espinosa de Los Reyes (Mexico City, Mexico). The rats were housed in a standard environment (temperature: 22 ± 1 °C; relative humidity: 55 ± 5%) with ad libitum access to food and water. After two weeks of acclimatization, breeding was conducted by placing two females with a male for 24 h. Gestational day (GD) 1 was marked by performing a vaginal smear to confirm the presence of sperm. The delivery day was appointed as postnatal day (PND) 0. The subjects of the study were two male offspring rats by litter. The experimental protocol was conducted following the Official Mexican Standard (NOM-062-ZOO-1999) and with the approval of the Institutional Committee for the Care and Use of Laboratory Animals of the Research (Mexico, Project 2017-3-138). We took special care to minimize animal suffering and the number of animals used. All experiments were conducted between 10:00 and 15:00, and independent groups were used.

### 2.3. Experimental Groups

In GD 1, the dams were divided into 4 groups of 6 as follows: (1) Control (CTL)-VEH group, (2) CTL-RSV group, (3) PS-VEH group, and (4) PS-RSV group. PS groups (VEH and RESV) were subjected to the method of restriction of movement. RSV groups were treated orally with 50 mg/kg BW/day of RESV, started on GD 1, and continued until GD 20 at 9 am daily. After birth, the offspring were sexed. Only male offspring were selected because the effect of PS on anxiety-like behavior is contradictory between male and female rats. In this sense, it has been reported that the estrus cycle stage influences the percentage of time the rats spend in the open arms of the elevated plus maze [21].

To avoid the litter effect and ensure the reliability of our study, we strictly adhered to the experimental designs for the study of PS published by Frasch et al. [22] and Benmhammed et al. [23], as well as the statistical design recommended by Festing [24] and Holson and Pearce [25]. Then, on PND 21, two male offspring were randomly selected from each litter, were weaned, and grouped into four groups according to their respective treatments. The male offspring were left undisturbed until early adulthood; by this time, 1 to 3 males became ill and died in each group. They underwent behavioral assessments: an open field test at PND 100, an elevated plus maze at PND 102, and a forced swim test at PND 105.

### 2.4. Model of Restraint of Movement

The dams (PS groups) were submitted to the method of restriction of movement, which consists of placing the dams in a well-ventilated acrylic tube, 7 cm wide and 19 cm long; this setup is designed for pregnant rats who cannot move and do not cause any harm. Therefore, it was conducted in 45 min periods, 3 times a day, from GD5 to GD20. In addition to this stressor, the dams were exposed to an artificial white light induced by a lamp during the last 24 h before birth [26,27]. The CTL group did not receive any intervention. The whole experimental process is shown in Figure 1.

### 2.5. Open Field Test

The open field test was used to evaluate the locomotor activity of the male offspring rats at PND 100 [28,29]. The test was conducted in a black acrylic box (60 × 60 × 60 cm) with divisions on the floor (15 equal squares). The results were expressed as the number of crossings and rearing over 5 min. The session was video recorded, and at the end, 5% ethanol was used to clean the surface.

### 2.6. Elevate Plus Maze

The elevated plus maze test was performed on PND day 102 day using a standard procedure [30]. A decrease in the time spent in and the number of entries into open arms is thought to be associated with increased anxiety-like behavior in rodents. The apparatus consisted of a cross-shaped platform having two opposing open arms (each 50 cm long × 10 cm wide), two opposing closed arms (each 50 cm long × 10 cm × 40 cm high), and a central arena (10 cm × 10 cm). The apparatus was supported by a platform that kept it 50 cm from the floor. Each rat was placed on the central platform facing one of the open arms and allowed to freely explore for 5 min. The apparatus was washed with 5% ethanol after each mouse to avoid odor interference. The tests were video recorded, and the following behaviors related to anxiety and stress were evaluated: time spent in open and closed arms (when more than 80% of the body is in one arm), rearing (rising to touch or not touch the wall of the maze), and head dipping (poking the head beyond the edge of the maze).

### 2.7. Forced Swimming Test

To further evaluate depression-like behavior, a forced swimming test was used at PND 105 [31]. This test was performed in an acrylic cylinder (46 cm in height, 20 cm in diameter) filled with water at 25 ± 1 °C to a depth of 30 cm, which ensures that the rat does not have the possibility of touching the bottom with its tail. Between every test, the water was changed to avoid any contamination. The test includes two sections: a 15 min pretest and a 5 min test 24 h later. The data were analyzed by an experimented researcher on a screen focusing on the behaviors related to stress and depression: swimming (when rats made horizontal movements), immersions, floating (when only performing the necessary movements to stay afloat), and climbing (when the rats were in vertical motion).

### 2.8. Tissue Preparation

All rats were euthanized on PND 110. The hippocampus and prefrontal cortex were quickly dissected from the brain and stored in a −80 °C refrigerator. When needed, four samples were randomly selected for biochemical analyses.

### 2.9. RNA Isolation

Total RNA was purified with an AllPrep DNA/RNA Mini kit (QIAGEN, Germantown, MD, USA), following manufacturer instructions. RNA integrity was analyzed by agarose gel electrophoresis, and its purity and quantity were obtained using a Multiskan spectrophotometer (Thermo Fisher Scientific, Waltham, MA, USA).

### 2.10. Real-Time Quantitative PCR

cDNA synthesis was conducted using the M-MLV reverse transcriptase enzyme (Invitrogen, Carlsbad, CA, USA) and 10 mM of each deoxynucleotide (Invitrogen), as specified by the supplier. cDNA was subjected to real-time quantitative PCR (RT-qPCR) using primers targeting *Bdnf* exon. *Gapdh* was used as an internal control of constitutive expression. The sequences of the specific primers are listed in Table 1. SYBR Green Master Mix (Thermo Fisher Scientific) was used as the detection method in a CFX96 Touch Real-Time PCR thermocycler (Bio-Rad, Hercules, CA, USA) following cycling conditions specified by the manufacturer. Relative quantification was performed with the ΔΔCt method [32].

### 2.11. Statistical Analyses

All statistical analyses were performed using GraphPad Prism version 10.3.1 (GraphPad, San Diego, CA, USA). The data are expressed as the means ± SEM, and *p*-value < 0.05 was considered statistically significant. The normal distribution of the data was evaluated by the Shapiro–Wilk test. All data were analyzed using two-way analysis of variance (ANOVA) with PS and RESV as independent variables followed by Bonferroni post hoc multiple comparison tests. To identify whether there is a relationship between the behavioral measures and *Bndf* transcripts expression, correlations were performed.

## 3. Results

### 3.1. Litter Size and Body Weight of Pups

The litters had a similar number of pups and males and females across all experimental groups. As shown in Table 2, there was no significant difference in the body weight of newborn pups. However, by weaning, the PS-VEH group showed less weight gain than the CTL-VEH group (*p* < 0.001). This effect was mitigated in the CTL-RESV and PS-RESV groups (*p* < 0.0001).

### 3.2. Open Field Activity

As shown in Figure 2a, the number of crossings showed a main effect of RSV [F(1, 36) = 11.51; *p* = 0.002] but no effect of PS [F(1, 36) = 2.28; *p* = 0.61], nor an interaction between these factors [F(1, 36) = 1.98; *p* = 0.17]. The post hoc test showed that RSV increased the number of crossings (*p* = 0.01).

The main effect of PS on rearing activity was significant [F(1, 36) = 6.95; *p* = 0.01], showing a decrease in the number of rearings. However, there was no significant effect of RSV [F(1, 36) = 0.32; *p* = 0.57] nor an interaction between these factors [F(1, 36) = 1.01; *p* = 0.32]. The post hoc test further confirmed the decrease in rearing activity due to PS (*p* = 0.01) (Figure 2b).

### 3.3. Elevated Plus Maze

The analysis of elevated plus maze data is presented in Figure 3. In total entries, there was a main effect of PS [F(1, 36) = 4.95; *p* = 0.033] and an interaction between factors [F(1, 36) = 6.62; *p* = 0.015] but no effect of RSV [F(1, 36) = 2.21; *p* = 0.147]). Post hoc testing revealed that PS decreased the total entries (*p* = 0.001), and this reduction was prevented by RSV (*p* = 0.009) (Figure 3a).

Rearing (Figure 3b): There was a main effect of RSV [F(1, 36) = 4.76; *p* = 0.037] and of the interaction between factors [F(1, 36) = 4.46; *p* = 0.043] but no effects of PS [F(1, 36) = 0.678; *p* = 0.416]. The post hoc test showed that the PS decreased rearing activity compared to the CTL groups (*p* = 0.040), and RSV prevented this effect (*p* = 0.005).

Head dips (Figure 3c): There was a main effect of PS [F(1, 36) = 4.17; *p* = 0.049] but no effect of RSV [F(1, 36) = 0.205; *p* = 0.654] or in the interaction between factors [F(1, 36) = 1.53; *p* = 0.224]. The Bonferroni test showed that PS decreased the number of head dips compared to the CTL groups (*p* = 0.023).

Open arm entries (number [n]): there was a main effect of PS [F(1, 36) = 8.46; *p* = 0.007] and in the interaction between factors [F(1, 32) = 5.77; *p* = 0.023] but no effect of RSV [F(1, 36) = 3.44; *p* = 0.073]. The Bonferroni test showed that PS decreased the number of open-arm entries compared to the CTL groups (*p* < 0.001), and this effect was prevented by RSV (*p* = 0.005) (Figure 3d).

Open arm duration (%): Analysis of data showed that there was a main effect in both PS [F(1, 36) = 14.45; *p* < 0.001] and RSV [F(1, 36) = 7.31; *p* = 0.011], as well as in the interaction between factors [F(1, 36) = 10.04; *p* = 0.003]. The post hoc test showed that the PS decreased the % of the open arm duration compared to the CTL groups (*p* < 0.001), and RSV prevented this effect (*p* < 0.001) (Figure 3e).

Closed arm entries (n): There was an interaction between factors [F(1, 36) = 6.89; *p* = 0.013], but no effect of PS [F(1, 36) = 1.42; *p* = 0.242] or RSV [F(1, 36) = 1.52; *p* = 0.227]. The Bonferroni test showed that PS decreased the number of closed-arm entries compared to CTL groups (*p* = 0.091), and this effect was prevented by RSV (*p* = 0.01) (Figure 3f).

Closed arm duration (%): Analysis of data showed that there was a main effect in both PS [F(1, 36) = 12.44; *p* = 0.001] and RSV [F(1, 36) = 8.79; *p* = 0.006] as well as in the interaction between factors [F(1, 36) = 8.71; *p* = 0.006]. The post hoc test showed that the PS increasing % closed arm duration compared to the CTL groups (*p* < 0.001), and RSV prevented this effect (*p* < 0.001) (Figure 3g).

### 3.4. Forced Swimming Test

Figure 4 presents the results of the forced swimming test. The analysis of swimming behavior revealed significant effects of RSV [F(1, 36) = 28.14; *p* < 0.001] but no effect of PS [F(1, 36) = 0.04, *p* = 0.845], or interaction [F(1, 36) = 0.156; *p* = 0.695]. The Bonferroni test showed that RSV groups spent more time swimming than the VEH groups (*p* < 0.01) (Figure 4a).

There was a significant effect of both the PS [F(1, 36) = 48.11; *p* < 0.001] and RSV [F(1, 36) = 117.22; *p* < 0.001] on immobility time. However, the two factors had no interaction effect [F(1, 36) = 0.009; *p* = 0.927]. The Bonferroni test revealed that PS rats exhibited significantly greater immobility time compared to CTL rats (*p* < 0.001), and RSV was effective in preventing this increase (*p* < 0.001) (Figure 4b).

The study found a significant main effect of PS [F(1, 36) = 29.59; *p* < 0.001] on climbing behavior. Similarly, there was a significant main effect of RSV [F(1, 36) = 31.62; *p* < 0.001]. However, there was no interaction between these factors [F(1, 36) = 0.07; *p* = 0.793]. The post hoc test showed that PS groups spent less time climbing than the CTL groups (*p* < 0.001), and RSV prevented this effect (*p* < 0.001) (Figure 4c).

### 3.5. Gene Expression of Bdnf

We investigated the prefrontal cortex and hippocampal gene expression of main *Bdnf* transcripts (exon IV, exon VI, and exon IX) to find molecular changes underlying PS and RSV effects (Figure 5).

Gene expression of *Bdnf* transcripts was evaluated in the prefrontal cortex, which showed a main effect of PS, RSV, and the PS x RSV interaction in exons VI and IX. By exon IV, there was only a main effect in RESV and the PS x RSV interaction (Figure 5a–c). The post hoc test showed an increase in expression levels in the PS-RSV groups compared to the PS-VEH groups (*p* < 0.0001) and the CTL-RESV groups (*p* < 0.001). Exon IV, main effect of PS: F(1, 12) = 4.57, *p* = 0.054; main effect of RSV: F(1, 12) = 7.62, *p* = 0.018; PS x RSV interaction: F(1, 12) = 13.06, *p* = 0.003]. Exon VI, main effect of PS: F(1, 12) = 15.63, *p* = 0.002; main effect of RSV: F(1, 12) = 18.63, *p* = 0.001; main effect of PS x RSV interaction: F(1, 12) = 23.02, *p* < 0.001. Exon IX, main effect of PS: F(1, 12) = 26.40, *p* < 0.001; main effect of RSV: F(1, 12) = 27.29, *p* < 0.001; PS x RSV interaction: F(1, 12) = 34.50, *p* < 0.001.

The gene expression profile in the hippocampus revealed that PS significantly affected the expression level of all *Bdnf* transcripts (Figure 5d–f). The main effect of RESV and the interaction of PS x RSV were only seen in exon IX. The Bonferroni test showed that the expression levels of all *Bndf* transcripts were higher in the PS-RSV groups compared to the CTL-RSV groups (*p* < 0.001). Additionally, there was an increase in the expression levels of exon IV and a decrease in exon IX in the PS-VEH groups compared to the CTL-VEH groups. Exon IV, main effect of PS: F(1, 12) = 44.50, *p* < 0.0001; main effect of RSV: F(1, 12) = 0.178, *p* = 0.687; PS x RSV interaction: F(1, 12) = 1.57, *p* = 0.234. Exon VI, main effect of PS: F(1, 12) = 12.57, *p* = 0.004; main effect of RSV: F(1, 12) = 0.856, *p* = 0.373; PS x RSV interaction: F(1, 12) = 0.736, *p* = 0.407. Exon IX, main effect of PS: F(1, 12) = 208.5, *p* < 0.0001; main effect of RSV: F(1, 12) = 385.7, *p* < 0.0001; PS x RSV interaction: F(1, 12) = 281.6, *p* < 0.0001.

### 3.6. Molecular and Behavioral Correlations

We performed linear regression analyses across all groups to determine the relationship between the expression of *Bdnf* transcripts and the effects of RESV on the behavioral data of each rat selected for biochemical analyses (Appendix A). The analysis of the overall regression showed only a significant correlation between swimming time and the expression of exons IV (r^2^ = 0.551; F(1, 15) = 17.15; *p* = 0.001) and IX (r^2^ = 0.437; F(1, 15) = 10.88 *p* = 0.005) in the hippocampus and exons IV (r^2^ = 0.380; F(1, 15) = 8.57; *p* = 0.011), VI (r^2^ = 0.469; F(1, 15) = 12.37; *p* = 0.003), and IX (r^2^ = 0.438; F(1, 15) = 10.89; *p* = 0.005) in the prefrontal cortex. This finding suggests a direct association between *Bdnf* transcripts expression and swimming time.

## 4. Discussion

In this study, we explored the effect of maternal RESV supplementation in prenatally stressed male offspring rats. The results suggest that PS throughout the gestational period led to reduced body weight gain and, in adulthood, increased anxiety- and depression-like behaviors and changes in *Bdnf* mRNA expression. Maternal RESV supplementation attenuated all these effects.

In recent years, a growing body of evidence suggests a connection between the early environment and later psychiatric disorders. Such research has been primarily inspired by the developmental origins of the health and disease (DOHaD) model, which proposes a link between fetal development and cardiovascular and metabolic disease in later life [33]. Barker and colleagues applied the DOHaD model early to mental health outcomes [34]. They investigated adult suicide rates of birth weight and growth during the first year of life. Their results showed that while birth weight alone was not a predictor, the average weight of 12-month-old infants was over 400 g lower in those who later died by suicide. These observations suggested that altered programming could influence growth in infancy and mood throughout life.

PS caused by maternal movement restraint is a well-established model of early stress recognized for causing enduring physiological, neurobiological, and behavioral changes [35]. Low birth weight is one of the most reported effects of gestational stress in newborn humans, and it has been associated with the development of a variety of metabolic diseases in adulthood [33]. On the other hand, most studies examining the bodyweight effects in rat offspring have reported contradictory findings. Therefore, in the current study, male offspring did not display an effect of the gestational stress manipulation on weight at PD1. A difference in weight was seen until the weaning (male offspring from stressed mothers weighing less than those from control mothers). Consistent with our findings that weight differences in offspring rats become noticeable only later in their development, Baker et al. [36,37] reported in two separate studies that gestational stress did not affect the weight of offspring from stressed mothers between postnatal days 2 and 24. However, weight differences appeared in female offspring from stressed mothers starting around 36 days of age, and a similar pattern was observed in male offspring from stressed mothers, but only in adulthood. This weight effect could be explained by abnormal secretion of growth hormone and irregularities in the HPA, hypothalamic–adrenal, and hypothalamic–thyroid axes [38,39]. Nevertheless, due to variations in results, it is crucial to consider the timing, dosage, and duration of the gestational stress model used [8].

Early-life exposure to adverse environments is often linked to higher rates of mood disorders in adulthood. Maternal movement restraint can lead to hyperactivation of the HPA axis, increased anxiety- and depression-related behavior, and changes in the expression of serotoninergic, dopaminergic, and glutaminergic receptors in the brains of offspring [21,35,40,41]. Studies have demonstrated that maternal movement restraint can exacerbate depressive-like behaviors in juvenile and adult male Wistar or Sprague Dawley rats, as observed in the forced swimming test and anxiety-like behaviors in adolescent male Sprague Dawley rats, as determined in the elevated plus maze [21,42,43]. Additionally, it is associated with a lower tendency to explore in the open field test [21,41]. Our results agree with these findings, as adult male rat offspring exposed to PS showed fewer rearings and crossings in the open field test compared to the CTL groups, which suggests that the PS group showed a lower tendency to explore. These rats also displayed anxiety-like behaviors, evidenced by reduced time spent in and number of entries into the open arms in the elevated plus maze test, and depression-like behaviors, indicated by increased immobility time and decreased climbing behavior in the forced swimming test. Several studies have shown that maternal stress can affect the development of central monoaminergic and glutamatergic neurons [44,45]. For instance, the offspring of stressed rats displayed hyperactive behavior in the central serotonin system, which could be linked to the anxiety-inducing effects of maternal movement restraint. Additionally, it has been proven that restraint stress exerted on the pregnant dam can have long-term effects on the dopaminergic system development in their offspring [40]. According to these studies, PS increases dopamine (DA) D2 receptors in limbic areas, reduces DA-stimulated release in cortical areas, increases levels in the nucleus accumbens, and disrupts the DA-glutamate balance, which could lead to an increased susceptibility to depression in the offspring.

BDNF is a member of the family of neurotrophins that are needed for the proper development and survival of neurons, and the dysregulation of its expression is related to neurodegenerative and neuropsychiatric disorders [46]. In rodent brains, the *Bdnf* gene contains at least eight 5′ exons (named from I to VIII) with independent promoters and one 3′ exon (IX), which encodes the BDNF protein [47]. *Bdnf* transcripts could be detected in nearly all brain regions, but exon IV is the most prevalent isoform [47,48]. Among all *Bdnf* transcripts/promoters, the regulatory mechanisms of promoters I to VI and a functional role have been demonstrated [49]. For example, different studies have suggested a relationship between promotors I, II, and IV and the response to antidepressants [50,51,52]; promotor IV has been related to anxiety-like behavior [53], addiction [54], and memory [55]; promotor VI has a potential role in dendritic structural plasticity and schizophrenia-like phenotypes [56]. In addition, using *Bdnf*-e4^−/−^ mice, it has been demonstrated that promoter IV is involved in regulating monoamine and cholinergic systems [57].

Because *Bdnf* has a very complex gene structure and can produce a variety of mRNAs with different functions, we decided to evaluate the effect of PS on the expression of different exons of *Bdnf* in brain areas involved in anxiety and depression, like the prefrontal cortex and hippocampus. Regarding affective disorders, it has been documented that the rodent *Bdnf* gene is regulated by stress and HPA axis activation [58], and an alteration of specific *Bdnf* transcripts has been reported in neuropsychiatric disorders [49]. Our data showed that PS increases the expression of exon IV while it reduces the expression of exon IX selectively in the hippocampus. In the prefrontal cortex, the prenatal manipulation did not affect the expression level of any of the exons evaluated. Recent animal studies have shown a wide variety of findings, and now there are opposing results on the effects of PS on *Bdnf* expression [9]. Although it has reported a decrease in total *Bdnf* mRNA and no changes or decreases in exon IV expression (for review, see [9]), our findings support those previously reported by Boulle et al. [59], which described increases in *Bdnf* exon IV expression in the hippocampus due to PS exposure. Similarly, our findings support the previously reported association between selective deficiency in promoter IV-dependent expression of *Bdnf* and depression-like behavior [53] and the increase in exons I and IV in the hippocampus associated with reduced anxiety-like behavior [60]. Considering the critical roles in the modulation level of total *Bdnf* mRNA and their exons by PS, our results emphasize the impact of stressful experiences in the prenatal stage on neurodevelopment and the origins of neuropsychiatric disorders.

Several beneficial effects of RESV have been reported, and many are particularly relevant to behavioral disorders. Its ability to regulate key molecular targets, including genes such as *Bdnf*, has been a subject of intense research [61]. Preclinical studies have shown that RESV plays an antidepressant and anxiolytic role in rodent models of anxiety and depression induced by estrogen deficiency, social isolation, and maternal separation [18,62,63]. Our findings suggested that RESV improved anxiety-like behaviors caused by maternal movement restraint, as evidenced by the significant increase in the time spent and the number of entries into the open arms in the elevated plus maze. Furthermore, RESV reduced the immobility time and increased swimming and climbing behaviors in the forced swimming tests, implying that RESV ameliorated depression-like behaviors caused by PS. In addition, our data showed that RESV increased the expression of *Bdnf* exons IV, VI, and IX in the prefrontal cortex and hippocampus of prenatally stressed adult male offspring. These results agree with the published data on the increasing effects of RSV on the expression of hippocampal levels of total *Bdnf* mRNA in male adult rats and *Bdnf* transcripts in pregnant rats [64,65].

Although we did not find a correlation between the expression of *Bndf* transcripts and immobility time in the forced swim test, the positive correlation with swimming time is related to the potentiation of active coping strategies, an effect seen with antidepressant drugs [66]. No significant correlations were found between *Bdnf* transcript expression and other behavioral measures, suggesting that RSV may be regulating the expression of other genes related to its anti-inflammatory and antioxidant activities [18].

In summary, the results of this study suggest that PS can lead to anxiety- and depression-like behavior in male offspring, effects being mediated by the regulation of *Bdnf* mRNA expression in the hippocampus. Additionally, RESV mitigated the anxiety- and depression-like behaviors induced by PS by increasing the expression of *Bdnf* exons IV, VI, and IX. This study is one of the first to evaluate the potential of pharmacological modulation of individual *Bdnf* transcripts by RSV for the treatment of anxiety and depression. However, it is crucial to note the limitations of our study. We did not provide information on the BDNF protein or whether changes in the expression of Bdnf transcripts are related to altered epigenetic mechanisms. Furthermore, we did not provide information on the effects of prenatal stress on female offspring. These should be addressed in future experiments conducted to provide new insights and a more comprehensive understanding of the effects of PS on both male and female offspring.

## 5. Conclusions

In conclusion, the results of this study suggested that PS can lead to anxiety- and depression-like behavior in male offspring, effects that are mediated by the regulation of *Bdnf* mRNA expression in the hippocampus. Additionally, RESV ameliorated anxiety- and depression-like behaviors induced by PS by increasing the expression of *Bdnf* exons IV, VI, and IX. While our study offers the possibility of novel and more specific treatments, future investigations are required to fully understand the mechanisms underlying the selective expression of different *Bdnf* transcripts in different brain areas and the temporal regulation of the expression of *Bdnf*.

## Figures and Tables

**Figure 1 brainsci-15-00210-f001:**
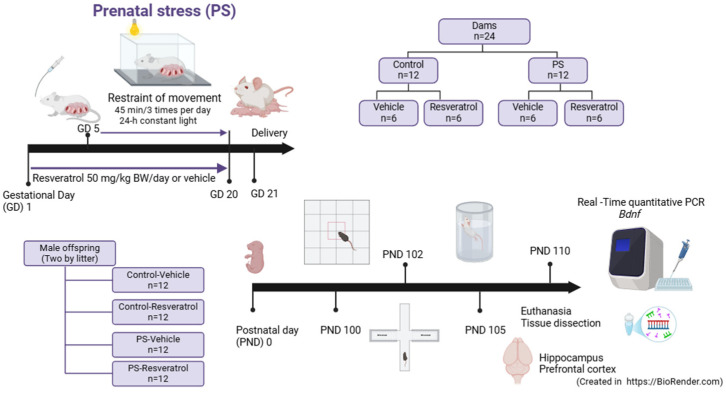
Experimental protocol. This study used restraint of movement to establish a prenatal stress model. Twenty-four pregnant rats were divided into 4 groups of 6 as follows: (1) Control (CTL)-VEH group, (2) CTL-RSV group, (3) PS-VEH group, and (4) PS-RSV group. On PND 21, two male offspring were taken from each litter, were weaned, and grouped into four groups according to their respective treatments. The male offspring were left undisturbed until early adulthood. They underwent behavioral assessments: an open field test at PND 100, an elevated plus maze at PND 102, and a forced swim test at PND 105. The offspring’s cerebral tissue was collected PND 110, and the expression of *Bdnf* exons was measured using RT-qPCR. (Created in https://BioRender.com).

**Figure 2 brainsci-15-00210-f002:**
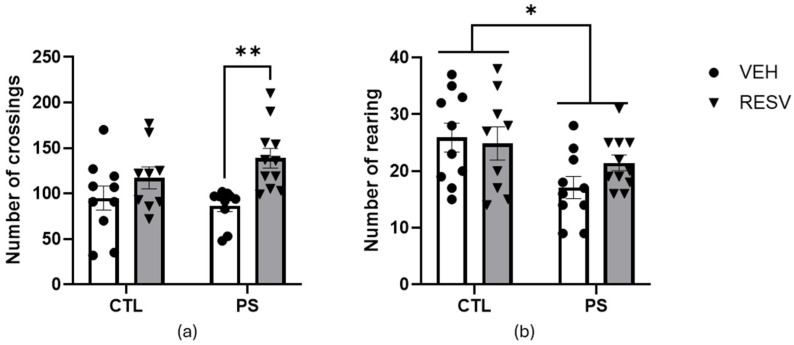
Effects of maternal resveratrol supplementation in the open field test of prenatal stress adult male offspring. (**a**) Number of crossings; (**b**) Number of rearing. All data are mean ± SEM with 10 animals in CTL-VEH and -RESV groups, 9 animals in PS-VEH group, and 11 animals in PS-RESV group. * *p* < 0.05; ** *p* < 0.01 by two-way ANOVA followed by Bonferroni post hoc tests. CTL = control; PS = prenatal stress; VEH = vehicle; RESV = resveratrol.

**Figure 3 brainsci-15-00210-f003:**
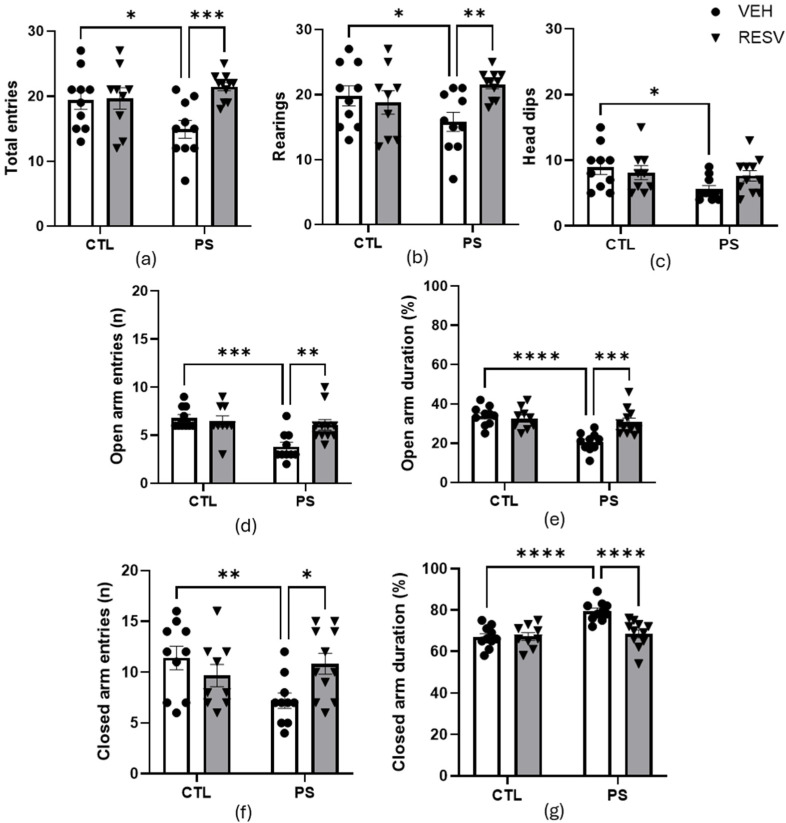
Effects of maternal resveratrol supplementation in the elevated plus maze of prenatal stress adult male offspring. (**a**) Total entries; (**b**) Rearings; (**c**) Head dips; (**d**) Open arm entries; (**e**) Open arm duration; (**f**) Closed arm entries; (**g**) Closed arm duration. All data are mean ± SEM with 10 animals in CTL-VEH and -RESV groups, 9 animals in PS-VEH group, and 11 animals in PS-RESV group. * *p* < 0.05; ** *p* < 0.01; *** *p* < 0.001; **** *p* < 0.0001 by two-way ANOVA followed by Bonferroni post hoc tests. CTL = control; PS = prenatal stress; VEH = vehicle; RESV = resveratrol.

**Figure 4 brainsci-15-00210-f004:**
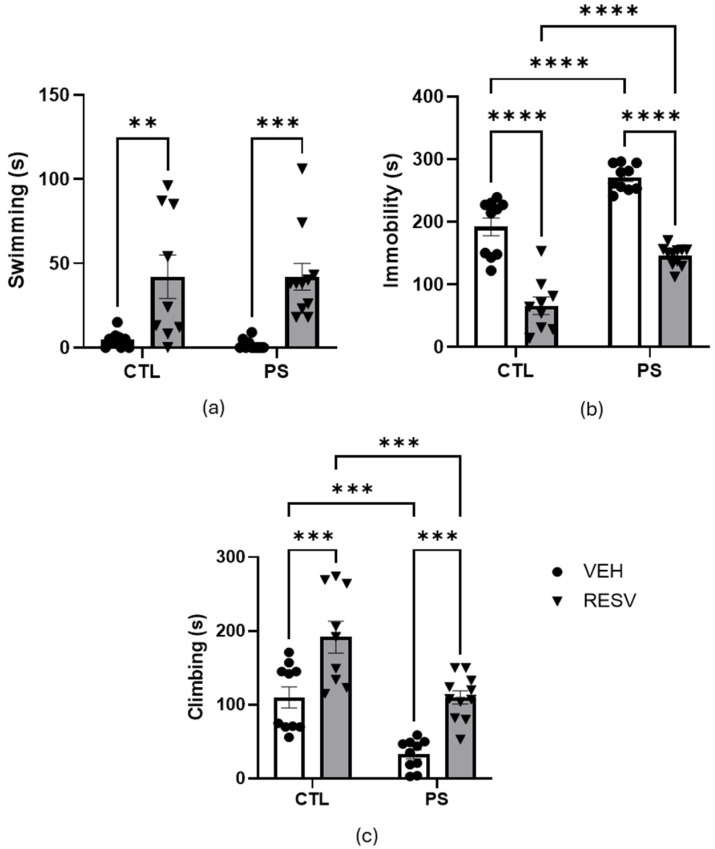
Effects of maternal resveratrol supplementation in the forced swim test of prenatal stress adult male offspring. (**a**) Swimming; (**b**) Immobility; (**c**) Climbing. All data are mean ± SEM with 10 animals in CTL-VEH and -RESV groups, 9 animals in PS-VEH group, and 11 animals in PS-RESV group. ** *p* < 0.01; *** *p* < 0.001; **** *p* < 0.0001 by two-way ANOVA followed by Bonferroni post hoc tests. CTL = control; PS = prenatal stress; VEH = vehicle; RESV = resveratrol.

**Figure 5 brainsci-15-00210-f005:**
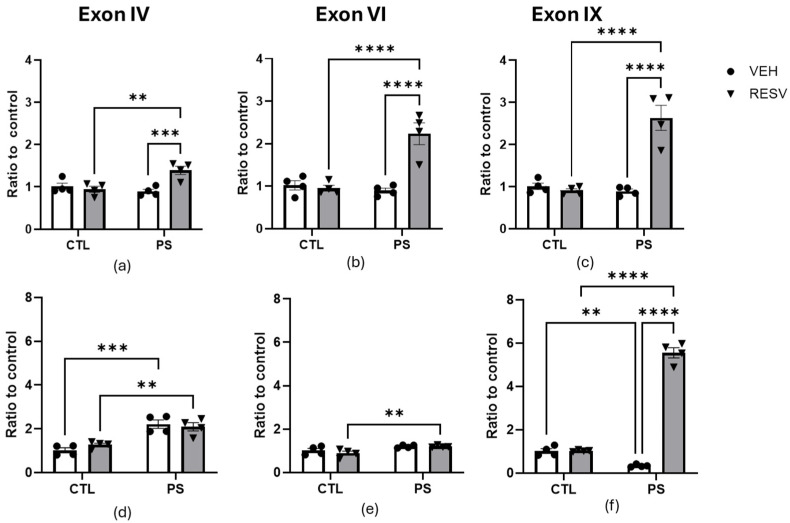
Effects of maternal supplementation of resveratrol on the expression of exons IV, IV, and IX of the *Bdnf* in the prefrontal cortex (**a**–**c**) and hippocampus (**d**–**f**) of prenatal stress adult male offspring. All data are mean ± SEM with 4 animals in each group. ** *p* < 0.01; *** *p* < 0.001; **** *p* < 0.0001 by two-way ANOVA followed by Bonferroni post hoc tests. CTL = control; PS = prenatal stress; VEH = vehicle; RESV = resveratrol.

**Table 1 brainsci-15-00210-t001:** Sequences of primers used for qRT-PCR amplification of transcripts of interest.

Gene Name	Primer Forward (5′ to 3′)	Primer Reverse (5′ to 3′)
*Bdnf* exon IV	TGGTGGCCGATATGTACTCC	ACTGAAGGCGTGCGAGTATT
*Bdnf* exon VI	TTGTTGTCACGCTCCTGGTC	GATGAGACCGGGTTCCCTCA
*Bdnf* exon IX	TTCCTCCAGCAGAAAGAGCA	TCCCTGGCTGACACTTTTGA
*Gapdh*	GGATGCAGGGATGATGTTC	TGCACCACCAACTGCTTAG

**Table 2 brainsci-15-00210-t002:** Summarized data on the litter size, number of males and females born, body weights, and changes in body weight.

Groups of Dams	Litter Size (Mean ± SEM)	No. of Pups Born	No. of Males and Females	Body Weight of Pups (g)	Changes i1n Body Weight (g)
CTL-VEH	12.00 ± 0.16	11 to 13	M = 4.66 ± 0.40 F = 7.33 ± 0.12	PND1 = 7.00 ± 0.09 PND21= 48.50 ± 1.17	41.00 ± 1.20
CTL-RESV	10.33 ± 1.85	9 to 12	M = 4.66 ± 0.40 F = 5.66 ± 0.74	PND1 = 6.70 ± 0.11 PND21 = 47.40 ± 0.66	40.60 ± 0.67
PS-VEH	10.66 ± 0.10	10 to 11	M = 5.33 ± 0.14 F = 5.33 ± 0.14	PND1 = 6.60 ± 0.09 PND21 = 34.00 ± 0.57	27.50 ± 0.58 *
PS-RESV	12.66 ± 2.05	12 to 13	M = 4.33 ± 0.43 F = 8.33 ± 0.41	PND1 = 7.08 ± 0.13 PND21 = 51.20 ± 0.94	44.30 ± 0.90

* *p* < 0.0001 vs. CTL-RESV and PS-RESV groups. CTL = control; VEH = vehicle; RESV = resveratrol; M = male; F = female; PND = postnatal day.

## Data Availability

The raw data of this article will be made available by the authors upon request.

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
