# Peer review of "Maternal Resveratrol Supplementation Attenuates Prenatal Stress Impacts on Anxiety- and Depressive-like Behaviors by Regulating Bdnf Transcripts Expression in the Brains of Adult Male Offspring Rats"

_brainsci, 2025, doi:10.3390/brainsci15020210_

Round 1

Reviewer 1 Report

Comments and Suggestions for Authors

1.    Why only males? Why would only male offspring be impacted by RESV exposure?

2.    Even though 2 pups per litter were pulled using litter as a factor in your statistics is still good practice.

See Golub & Sobin, 2020

3.    For all bar graphs please show individual animal dots.

4.    For all graphs, please report- N of each experimental group.

5.    In the discussion, please explain the exon differences and relevance. You write exon IV was increased but don’t describe how it was regulated in past anxiety and depression work. Just add a little more detail.

6.    Can you run any analysis that examines hippocampus exon expression and your specific behaviors?

Were animals that had increases in exon expression the ones with improved behaviors?

7.    If, RSV regulated BDNF in the PFC as well, how do you know where the behavioral benefits came from?

8.    50 mg per kg of RESV for dams, how would that relate to a human? How much RESV would a person have to consume to find a similar benefit?

Minor issues:

PS groups (define PS line 96, page 3) maybe define it on page 2 line 61

Author Response

1. Summary

Thank you very much for taking the time to review this manuscript. We have completed the revision of the manuscript according to your guidance. Please find the detailed responses below and the corresponding revisions or corrections highlighted in the re-submitted files.

2. Point-by-point response to Comments and Suggestions for Authors

Comments 1: Why only males? Why would only male offspring be impacted by RESV exposure?

Response 1: Thank you for pointing out the need to provide further clarifications on because we used only male offspring. We decided to test only male rats because data on the effect of PS on anxiety-like behavior are conflicting. It is our knowledge that the existence of sexual dimorphism in the outcome of PS has been hypothesized more than 30 years ago. There are reports of increased anxiety-like behavior in adult female, but not male, PS rats.  In addition, it was reported that the estrus cycle stage influenced the percentage of time the rats spent in the open arms of the elevated plus maze. However, due to variations in results, it is crucial to consider the timing, dosage, and duration of the gestational stress model used (Weinstock, 2017; doi: 10.1016/j.ynstr.2016.08.004).

Although our study's limitation is that it does not provide information on the effects of prenatal stress on female offspring, future experiments should be conducted to provide new insights and a more comprehensive understanding of the effects of PS on both male and female offspring.

Comments 2: Even though 2 pups per litter were pulled using litter as a factor in your statistics is still good practice. See Golub & Sobin, 2020.        

Response 2: Thank you for pointing out the need to provide further clarifications on “litter effect”. Appropriate handling of litter is only one consideration of experimental design and statistical analysis that leads to valid, reproducible data. For the experimental design, we tried to follow the recommendations of the protocols published by Frasch et al. (2018; doi: 10.1007/978-1-4939-7828-1_19) and Benmhammed et al. (2019; doi: 10.1007/978-1-4939-9554-7_10), as well as the recommendations for the statistical design for studies in animal model development published by Festing (2006; doi: 10.1093/ilar.47.1.5).

Golub and Sobin (2020) propose a robust solution for handling litter effects in the mixed linear model. This model controls for litter clustering by partitioning litter variance from error variance, thereby reducing error variance and enhancing the power of F tests of the independent variable. However, it is important to consider additional factors in many behavioral tests, such as measuring performance across intervals, trials, and/or days.

Comments 3: For all bar graphs please show individual animal dots.

Response 3: All bar graphs were corrected and now they show the individual animal dots.

Comments 4: For all graphs, please report- N of each experimental group.

Response 4:     All graphs were corrected and now they report N of each experimental group.

Comments 5: In the discussion, please explain the exon differences and relevance. You write exon IV was increased but don’t describe how it was regulated in past anxiety and depression work. Just add a little more detail.

Response 5: We aggregate information in more detail in the discussion on the exon differences and relevance, as well as their role in anxiety and depression.

In the revised manuscript, please refer to page 11, lines 374-407.

Comments 6: Can you run any analysis that examines hippocampus exon expression and your specific behaviors? Were animals that had increases in exon expression the ones with improved behaviors? 

Response 6: We agree with this comment. Therefore, we performed linear regression analyses to determine the relationship between the expression of Bdnf transcripts and the effects of RESV on the behavioral data of each rat. The analysis of the overall regression can be seen in Table S1. In addition, we added a brief discussion.

In the revised manuscript, please refer to page 5, lines 186-187;  page 11, lines 308-315; and pages 11-12, lines 423-428.

Comments 7: If, RSV regulated BDNF in the PFC as well, how do you know where the behavioral benefits came from?

Response 7. This comment is related to the above one. Therefore, it could be responded to with linear regression analyses.

In the revised manuscript, please refer to page 5, lines 186-187; page 11, lines 308-315; and pages 11-12, lines 423-428.

Comments 8: 50 mg per kg of RESV for dams, how would that relate to a human? How much RESV would a person have to consume to find a similar benefit?

Response 8: Thank you for pointing out the need to provide further clarifications on dose used. The daily dose used in the present work was 50 mg/kg BW daily. This daily dose is between the range used in human studies. Using a previously defined formula (Reagan-Shaw et al.,2008; doi: 10.1096/fj.07-9574LSF), it can show that 50 mg/kg BW in the rat would correspond to 8.108 mg /kg in humans. For comparison, in clinical studies in which the actions of RESV on several inflammatory and metabolic markers were investigated (Brown et al., 2024; doi:10.3390/ijms25020747), the daily dose given to the patients was until 1 g, which corresponds to 14.3 mg/kg in patients weighing 70 kg.

Minor issues: PS groups (define PS line 96, page 3) maybe define it on page 2 line 61

Response: The abbreviature PS is defined on page 2 line 46.

Reviewer 2 Report

Comments and Suggestions for Authors

This research article is dealing with the effect of resveratrol on male offspring rats after prenatal stress induction in well-documented animal model. The experimental planning and setup is correct and the methods used are described well and are modern. The offspring animals are examined in adulthood concerning their behavior, like anxiety and depression. The CNS regions are examined according to these changes, looking at the expression level of BDNF. Resveratrol were administered to the dams, with or without stress situations. The authors claim, that resveratrol protected the offspring animals from psychological disorder development, with the mRNA expression level increase of BDNF. This is a very interesting question.

My questions: beside weight differences, are there any additional change in young animals of stressed mothers? What was the main reason that male offspring were chosen for experiments? In figure 4. both controls (without or with stress) are shown as 1, so I could not judge the statement, that stress has reduced the expression of BDNF. Similarly, the difference between the two brain regions cannot be seen (in the written numbers maybe, but it should be obvious on the graph, as well, or at least put into table, please). What is the reason of the examination of the three exons separately, and what is the consequence of the difference between them after the treatments? The epigenetic results are really missing, or any cell signaling pathway activation.

Author Response

1. Summary

Thank you very much for taking the time to review this manuscript. We have completed the revision of the manuscript according to your guidance. Please find the detailed responses below and the corresponding revisions or corrections highlighted in the re-submitted files.

2. Point-by-point response to Comments and Suggestions for Authors

Comments 1:  beside weight differences, are there any additional change in young animals of stressed mothers?

Response 1:  In the experimental design, we did not consider measuring any other physical or physiological variable in the offspring. We can only comment that it was the group where we had more sick animals.

Comments 2: What was the main reason that male offspring were chosen for experiments?

Response 2: Thank you for pointing out the need to provide further clarifications on because we used only male offspring. We decided to test only male rats because data on the effect of PS on anxiety-like behavior are conflicting. It is our knowledge that the existence of sexual dimorphism in the outcome of PS has been hypothesized more than 30 years ago. There are reports of increased anxiety-like behavior in adult female, but not male, PS rats.  In addition, it was reported that the estrus cycle stage influenced the percentage of time the rats spent in the open arms of the elevated plus maze. However, due to variations in results, it is crucial to consider the timing, dosage, and duration of the gestational stress model used (Weinstock, 2017; doi: 10.1016/j.ynstr.2016.08.004).

Although our study's limitation is that it does not provide information on the effects of prenatal stress on female offspring, future experiments should be conducted to provide new insights and a more comprehensive understanding of the effects of PS on both male and female offspring.

Comments 3: In figure 4. both controls (without or with stress) are shown as 1, so I could not judge the statement, that stress has reduced the expression of BDNF.

Response 2: The graphs in Figure 4 were corrected and now show the individual data.

Comments 4. Similarly, the difference between the two brain regions cannot be seen (in the written numbers maybe, but it should be obvious on the graph, as well, or at least put into table, please).

Response 4: The graphs in Figure 4 were corrected and now show the individual data.

Comments 5: What is the reason of the examination of the three exons separately, and what is the consequence of the difference between them after the treatments?

Response 5: We aggregate information in more detail in the discussion on the exon differences and relevance, as well as their role in anxiety and depression.

In the revised manuscript, please refer to page 11, lines 374-407.

Comments 6: The epigenetic results are really missing, or any cell signaling pathway activation.

Response 6: We agree with this comment. In the manuscript, we established that this is a limitation of our work and should be addressed in future experiments. Currently, we are working on epigenetic data.

Reviewer 3 Report

Comments and Suggestions for Authors The paper by Vera-Juárez et al.  is devoted to studying the possibility of correcting the negative consequences of prenatal stress by long-term treatment with Resveratrol ( RESV). To achieve this goal, female rats were subjected to both ROL treatment and chronic restriction stress during pregnancy. The authors demonstrated that the prenatal stress led to reduced weight gain and, in adulthood, increased anxiety- and depression-like behaviors and changes in Bdnf mRNA expression. However, these effects were attenuated by maternal RESV supplementation. The paper is well structured, the tables and figures complement the text well. Abstract gives all the necessary information about the contents of the paper, keywords are appropriately chosen. The reference list covers the relevant literature adequately (the authors cite 50 sources).    However, I have some remarks. 1) It would be better if the authors drew the design of the experiment in a figure. 2) In section 2.2. the authors write: "Forty female Wistar rats were provided...". At the same time, the authors write in section 2.3.: "... the dams were divided into four groups of 5..." . It turns out that only 20 females out of 40 were used in the experiment? 3) Why were only Bdnf exon IV, Bdnf exon VI and Bdnf exon IX transcripts assessed? This is not explained in the Introduction, Materials or Discussion. 4) Is it accurate that in Table 2 the Litter size (mean ± SEM) for the CTL-RESV group is 10.33 ± 18.85? 5) Gapdh is a poor choice of housekeeping gene because it has many pseudogenes. My advice is to try to avoid using it.

Author Response

Response to Reviewer 3 Comments

1. Summary

Thank you very much for taking the time to review this manuscript. We have completed the revision of the manuscript according to your guidance. Please find the detailed responses below and the corresponding revisions or corrections highlighted in the re-submitted files.

2. Point-by-point response to Comments and Suggestions for Authors

 Comments 1:   It would be better if the authors drew the design of the experiment in a figure.

Response 1:  A figure showing the design of the experiment was added.

Comments 2:   In section 2.2. the authors write: "Forty female Wistar rats were provided...". At the same time, the authors write in section 2.3.: "... the dams were divided into four groups of 5..." . It turns out that only 20 females out of 40 were used in the experiment?

Response 2:    Thank you. This error has been corrected. Twenty-four female Wistar rats were divided into four groups (6 each).

In the revised manuscript, please refer to page 2, line 82; page 3, line 96.

Comments 3:  Why were only Bdnf exon IV, Bdnf exon VI and Bdnf exon IX transcripts assessed? This is not explained in the Introduction, Materials or Discussion.

Response 3:    We aggregate information in more detail in the discussion on the exon differences and relevance, as well as their role in anxiety and depression.

In the revised manuscript, please refer to page 11, lines 374-407.

Comments 4: Is it accurate that in Table 2 the Litter size (mean ± SEM) for the CTL-RESV group is 10.33 ± 18.85?

Response 4:    Thank you. This error has been corrected (10.33±1.85)  

In the revised manuscript, please refer to page 5, Table 2

Comments 5:   Gapdh is a poor choice of housekeeping gene because it has many pseudogenes. My advice is to try to avoid using it.

Response 5:    Thanks for the suggestion. According to the design of the oligonucleotides, they were prevented from falling into pseudogenes, the alignment with the first blast was carried out, and it was verified that there were no pseudogenes. Also, when performing the experiments, it was verified that the levels of Gapdh did not vary between the replicate experiments and the conditions, which indicates that it was not a pseudogene.  Moreover, we are only using it as a reference to buy, and the most important thing is that it does not vary and that the expression remains stable, as in our case.

Round 2

Reviewer 2 Report

Comments and Suggestions for Authors

I have read the authors’ answers and looked at the modified manuscript. I accept the answers as well as the changes made and now accept the manuscript for publication.

Author Response

Response to Reviewer 2 Comments

Comments 1: I have read the authors’ answers and looked at the modified manuscript. I accept the answers as well as the changes made and now accept the manuscript for publication.

Response 1: Thank you very much for taking the time to review this manuscript. Your comments and suggestions helped us to improve it.
